# Affordable Processing of Edible Orthopterans Provides a Highly Nutritive Source of Food Ingredients

**DOI:** 10.3390/foods10010144

**Published:** 2021-01-12

**Authors:** Forkwa Tengweh Fombong, John Kinyuru, Jeremiah Ng’ang’a, Monica Ayieko, Chrysantus Mbi Tanga, Jozef Vanden Broeck, Mik Van Der Borght

**Affiliations:** 1Molecular Developmental Physiology and Signal Transduction lab, Division of Animal Physiology and Neurobiology, Department of Biology, KU Leuven, Naamsestraat 59 Box 2465, 3000 Leuven, Belgium; tengweh.fombong@kuleuven.be (F.T.F.); jozef.vandenbroeck@kuleuven.be (J.V.B.); 2Department of Food Science and Technology, Juja (Main) Campus, Jomo Kenyatta University of Agriculture and Technology, P.O. Box 62,000, Nairobi 00200, Kenya; jkinyuru@agr.jkuat.ac.ke (J.K.); kandojeremies@yahoo.com (J.N.); 3International Centre of Insect Physiology and Ecology (icipe), P.O. Box 30772, Nairobi 00100, Kenya; ctanga@icipe.org; 4Lab4Food, Department of Microbial & Molecular Systems, KU Leuven, Kleinhoefstraat 4, B-2440 Geel, Belgium; 5School of Agricultural and Food Science, Jaramogi Oginga Odinga University of Science and Technology, Bondo-Usenge Road, P.O. Box 210, Bondo 40601, Kenya; monica_ayieko@yahoo.com

**Keywords:** edible insects, East Africa, complementary foods, processing, protein quality, fat quality, micronutrients

## Abstract

Edible orthopterans (grasshoppers, crickets, and locusts) are major delicacies, especially across sub-Saharan Africa. Their promotion as food ingredients is increasingly gaining momentum. This study evaluates the nutritional profiles of three widely consumed orthopterans: *Gryllus bimaculatus*, *Locusta migratoria*, and *Schistocerca gregaria* after blanching and oven-drying. All three species had high protein (65.3, 54.2, and 61.4% on a dry matter (DM) basis for *G. bimaculatus*, *L. migratoria*, and *S. gregaria*, respectively) and fat contents. Oleic (22.9–40.8%) and palmitic (26.1–43.0%) were the two most abundant fatty acids. All essential amino acids (in mg/100 g protein) were present, with glutamic acid (120–131), alanine (90.2–123), and leucine (82.3–84.6) being the most abundant. The minerals (in mg/100 g dry matter) potassium (796–1309) and phosphorus (697–968) were moderately high, and iron (4.60–7.31), zinc (12.7–24.9), manganese (0.40–7.15), and copper (1.20–4.86) were also observed in the samples. Vitamin B_12_ contents were high (0.22–1.35 µg/100 g dry matter). Our findings demonstrate that the excellent nutritional profile of the three processed insects could serve as promising alternative ingredients for improving food and nutritional security.

## 1. Introduction

With a rapidly increasing world population, a declining availability of arable land, and a growing demand for sustainable food sources, insects are gaining ground as a promising alternative source of protein and other essential nutrients for improving human nutrition [1,2,3]. More than 2000 insect species are currently consumed globally by approximately two billion people [2,4]. Pioneered by the works of Meyer-Rochow [5], followed by DeFoliart [6] and then Van Huis [7], the UN’s Food and Agriculture Organization (FAO) now regards insects as a potentially sustainable food source that can be used to address some of the global food security concerns. This has stimulated the increasing use of insects in diets [8], thus generating worldwide interest in developing and using insect-food products. Insects are widely consumed in Asia, South America, and Africa, although it is not yet standard practice in Europe and North America [2,9]. In Africa, edible insects continue to play a significant role in nourishing indigenous communities, especially in sub-Saharan Africa, where the consumption of insects is an established cultural practice.

The desert locust, *Schistocerca gregaria* Forsskål, and the migratory locust, *Locusta migratoria* Linnaeus, are sporadic pests of historical importance in many countries across Africa, the Middle East, the Indo-Pakistan Peninsulas, and Europe, where they cause substantial crop losses during plagues [10,11]. They are commonly eaten as an important source of food by many marginalised communities [5,12,13]. For example, in Sudan, locusts are either eaten raw or prepared by boiling, frying, or sun-drying [13]. The most recent locust outbreak started in the horn of Africa in December 2019 [14], with countries in this region (Ethiopia, Kenya, Somalia, Uganda, and South Sudan) still battling their worst locust outbreak in decades [14]. Swarming locusts cause massive damage due to voracious feeding on crops, pastures, and any green vegetation, thereby significantly affecting the livelihoods of people. Studies have demonstrated that, in many countries, farmers forgo pesticidal control of locusts in favour of harvesting and selling them, generating much-needed income for their households compared to the sale of food crops.

Numerous Orthopteran species with pest statuses are consumed in South America and Africa. For example, in Mexico, massive hand-picking of the chapuline grasshoppers (*Sphenarium purpurascens)* that infested alfalfa fields played a significant role in decreasing environmental damage while generating an additional source of nutrition and income through sales for human consumption [15]. Several species of locusts are widely eaten in sub-Saharan Africa, including *S. gregaria*; *L. migratoria*; the red locust, *Nomadacris septemfasciata*; and the brown locust, *Locustana pardalina* [7]. Large-scale wild harvesting of locusts for sale as a protein-rich ingredient for human food or animal feed may significantly contribute to their control and help to reduce the application of chemical pesticides and its associated environmental pollution [16].

Currently, crickets are one of the most widely farmed insect groups for human consumption as food and/or high-quality protein ingredients for inclusion in livestock feeds in many regions of the world [17,18,19]. Crickets are a popular group of insects for this purpose for several reasons: an excellent feed conversion ratio, a short generation time, tolerance of high densities, being generalist feeders, broad disease tolerance, and not undergoing diapauses [19,20,21]. Thus, crickets are increasingly being considered a sustainable alternative to animal and plant protein sources, with crude protein values above 58% dry weight [21].

In Kenya, where there are currently large swarms of *S. gregaria*, beef is the most consumed protein source and retails between US$ 3 and 5 per kg. In this study, we hypothesise that the consumption of locusts can provide adequate or superior nutritional needs compared to conventional livestock and crop sources. Despite the potential economic importance of these insects, very little attention has so far been given to the nutrient content of locust species in East Africa, where outbreaks are likely to become highly frequent [22]. Also, evaluating the nutritional content of insects using low-cost and affordable processing techniques need particular attention. Blanching followed by oven-drying before milling into powders is a cheaper alternative delivering the same nutritional end-product in comparison to more costly processes such as freeze-drying [23]. Several parameters that confer the nutritional attributes of proteins and lipids have been described [24]. However, the paucity of empirical data for less-known protein quality attributes (such as protein efficiency ratio, nutritional index, and biological value) has inspired the use of other methods using several data-derived equations that can adequately predict these attributes [25,26,27]. Recently, predicted nutritional quality parameters derived from amino acid and fatty acids values had been used to predict the quality of proteins and lipids, respectively, in foods [27]. Previous studies have reported the content of various nutrients for *S. gregaria*, *L. migratoria*, and *G. bimaculatus* [21,28,29]. However, a complete nutritional breakdown has not been reported for any of these insects. In addition, no studies comparing the nutrient potential of these two pest locusts to that of their more commonly reared orthopteran counterpart, the cricket—*Gryllus bimaculatus*—exist. Finally, there are no reports that used predicted nutritional quality parameters to interpret the nutritional value of edible insects, including locusts and crickets. We, therefore, performed a comparative analysis of the complete nutritional profiles (proximate composition, amino acids, fatty acids, minerals, and vitamins) and nutritional quality parameters of all three species, *G. bimaculatus*, *L. migratoria*, and *S. gregaria*, after blanching and oven-drying.

## 2. Materials and Methods

### 2.1. Rearing and Maintenance of Insect Colonies

All three test species, *G. bimaculatus*, *L. migratoria*, and *S. gregaria*, were reared at KU Leuven’s Zoological Institute (Department of Biology, Leuven, Belgium). The long-term colony of *S. gregaria* originated from the *Aquazoo* in Düsseldorf, Germany, which had them from a wild population in Nigeria. This colony has been maintained at the Laboratory of Molecular Developmental Physiology and Signal Transduction (KU Leuven, Belgium) for 35 years, whereas *L. migratoria* original stock was purchased from the company *Sprinkhanenwinkel* in Someren (The Netherlands). Both locust species were reared under crowded conditions (>200 locusts/cage) in perplex glass cages of 50 cm × 50 cm × 100 cm. Incandescent light bulbs (40 W) suspended in the cages provided extra warmth and light. Plastic cylindrical containers with open tops (diameter of approximately 7.5 cm) containing sterile peat and sand were placed at the bottom of the cages for egg-laying. Vertical wire meshes were provided for extra perching space. Locusts were fed daily ad libitum with fresh cabbage leaves (*S. gregaria*) or ryegrass (*L. migratoria*), supplemented with dry oat flakes for both species. Two-spotted field crickets (*G. bimaculatus*) were purchased from a local pet shop and reared in small plastic boxes (50 cm × 50 cm × 50 cm) containing 50–60 crickets. Crickets were fed daily with dry fish flakes (sera^®^ GVG-Mix Nature) from a pet shop and with apples and carrots from local food vendors. Eggs were laid in moist double-layered cotton wool. The rearing room of the three insect species was maintained at a constant temperature of 30 °C ± 1 °C, relative humidity of 40–60%, and a photoperiod of 14 h light/10 h dark.

### 2.2. Sample Preparation

Approximately 1 kg of each locust was sampled one week into adulthood, and 1 kg of crickets was sampled at adult stages. All insect samples were sacrificed by blanching in distilled water at 100 °C for 4 min, before being cooled in ice water and drained. The insects were immediately dried using a laboratory oven (Memmert UF 110, Memmert, Schwabach, Germany) at 60 °C for 24 h. The oven-dried samples were ground into a fine powder using a laboratory blender (Camlab, Over, UK). Samples were stored at −20 °C for further analysis.

### 2.3. Nutritional Composition Parameters of the Three Insects

Total compositions were obtained for moisture, crude protein, crude fat, ash, and chitin (fibre) using the methods outlined in our previous study [23]. 

In brief, the moisture content was analysed using a forced-air oven (UF 110, Memmert, Schwabach, Germany) at 105 °C for 17 h.

The total organic nitrogen content was determined by the Kjeldahl method using a steam distillation apparatus (Vapodest 20, Gerhardt, Königswinter, Germany). The method was verified using acetanilide (Sigma-Aldrich, St. Louis, MO, USA) as the reference standard; the method blank was also included. Protein content was calculated with 6.25 as the conversion factor.

Crude fat content was measured using the Soxhlet method using petroleum ether (technical, boiling point 40–60 °C, VWR Chemicals, Fontenay-sous-Bois, France) as the solvent. The solvent was then removed using a rotary evaporator (Büchi, R-200, Essen, Germany) at 50 °C. The resulting oil extract was later used for fatty acid determination.

The ash contents were determined gravimetrically using a muffle furnace (B 180, Nabertherm, Lilienthal, Germany) until a constant weight was reached at 550 °C. The ashes obtained were collected and preserved in a low-density polyethylene container for mineral analysis.

The chitin content of the insect samples was measured gravimetrically after deproteinization using 1.0 M NaOH, AnalaR NORMAPUR and subsequent demineralization with 1.0 M HCl, TITRINORM.

The nitrogen-free extract (*w_NFE_*) was calculated as follows [30]:(1)$$w_{NFE}=100-(w_{proteins}+w_{lipids}+w_{ash}+w_{fibre}+w_{moisture})$$
where *w* = mass fraction (g/100 g dry matter)

The corresponding standard deviation (*SD*) of the *NFE* fraction, i.e., *SD**w_NFE_*, was calculated as follows:

(2)$$SDw_{NFE}=\sqrt{SDw_{proteins}{}^{2\;}+\;SDw_{lipids}{}^{2\;}+\;SDw_{ash}{}^{2\;}+\;SDw_{fibre}{}^{2\;}+\;SDw_{moisture}{}^{2\;}}$$

The energy content (*EC* in Kcal/100 g dry matter) was estimated using meat/fish-specific factors according to the guidelines of the FAO 2003 food nutrition paper report [30] using the following formula:(3)$$EC=w_{proteins}\times 4.27\mathrm{kcal}/g+w_{lipids}\times 9.02\mathrm{kcal}/g+w_{NFE}\times 3.87\mathrm{kcal}/g$$

Additionally, the corresponding standard deviation for energy content (*SD*_EC_) was calculated as follows:(4)$$SD_{\mathrm{EC}}=\sqrt{\frac{SDw_{proteins}}{w_{proteins}{}^{2}}+\frac{SDw_{lipids}}{w_{lipids}{}^{2}}+\frac{SDw_{NFE}}{w_{NFE}{}^{2}}}$$

#### 2.3.1. Amino Acid Analysis

Amino acid profiles were determined after acid and alkaline (for tryptophan) hydrolysis of the dried and defatted samples followed by separation and quantification using amino acid standards on a Waters Acquity UPLC H-class system, as described previously by Fombong et al. 2017 [23]. Briefly, approximately 25.00 mg of the samples were subjected to acid hydrolysis using 6.0 M HCl TITRINORM at 110 °C for 22 h in an inert atmosphere to prevent oxidation. It should be noted that asparagine and glutamine were converted into aspartic acid and glutamic acid, respectively, during the acid hydrolysis step. Tryptophan determination was achieved separately after hydrolysis with 4.0 M LiOH, EXTRAPUR for 22 h at 110 °C.

UPLC separation of these amino acids was performed on an Acquity UPLC (Waters, Milford, MA, USA), consisting of a photo diode array (PDA) detector, a column heater, a sample manager, a binary solvent delivery system, and an Acc•QTagTM Ultra column (2.1 i.d. × 100 mm; Waters, USA). Sample derivatization was achieved using the Waters AccQ•Tag Ultra Chemistry Package (Waters, Milford, MA, USA). Prior to derivatization, the sample was diluted to 1:10 ratio with 0.1 M HCl and excess acid was neutralised with 0.1 M NaOH, AnalaR NORMAPUR. Gradient elution was applied according to Waters AccQ•Tag Ultra method (AccQ•Tag Ultra Eluent A Concentrate (10-times diluted) (Waters); AccQ•Tag Ultra Eluent B (Waters)) at a flow rate of 0.7 mL·min^−1^, and the column temperature was maintained at 60 °C. For quantification of the amino acids in the sample, the system was calibrated using the analytical standards of amino acids (AAS18, Sigma-Aldrich, St. Louis, MO, USA). The data were processed using the Empower 2 software (Waters, Milford, MA, USA). Analyses of all samples were performed in triplicate.

#### 2.3.2. Fatty Acid Analysis

Fatty acid methyl esters were prepared from the three insect oil extracts obtained from the Soxhlet extraction, separated, and identified by gas chromatography coupled with mass spectrophotometry as described by Fombong et al. 2017 [23]. In brief, 0.025 g of extracts of the insect oils were weighed. Using 0.5 M sodium methoxide (VWR International BVBA, Leuven, Belgium) and 10% methanolic boron trifluoride (BF_3_) solution (Sigma-Aldrich, St. Louis, MO, USA), the acylglycerols and free fatty acids in the sample were (trans)esterified into fatty acid methyl esters (FAMEs); 1.0 µL of the esterified samples or fatty acid methyl ester standard (Sigma-Aldrich, St. Louis, MO, USA) was injected into the gas chromatograph (Agilent 7820A-5977E GC-MSD, Agilent Technologies, Santa Clara, CA, USA). Using methyl tricosanoate (Sigma Aldrich, St. Louis, MO, USA) as the internal standard, the MassHunte*r* software (Agilent Technologies, Santa Clara, CA, USA) was used to compute the concentrations of fatty acids in mg/l. Analyses of all samples were performed in triplicate.

#### 2.3.3. Mineral and Trace Elemental Analysis

The ash from ash analysis was used to estimate the mineral composition in all insect samples based on previous study by Fombong et al. 2017 [23]. Briefly, the ashes were dissolved in 65% HNO_3_ (VWR Chemicals, Fontenay-sous-Bois, France) and then diluted ten-fold to appropriate concentrations based on the mineral element and the resulting calibration curve. The calibration curves were obtained using standard solutions from certified stock solutions containing 1000 ppm of the elements investigated (Chem Lab, Zedelgem, Belgium). The contents of each investigated element (sodium, calcium, phosphorus, potassium, magnesium, iron, zinc, manganese, and copper) were determined by inductively coupled plasma optical emission spectrometry (ICP-OES) measurements (Optima 4300™ DV ICP-OES, Perkin Elmer, Wellesley, MA, USA). The parameter settings for the ICP-OES are outlined in Table 1 below. Data were processed using the WinLab 32 software. All samples were analysed in duplicate. For each element, three readings (replicates) were measured by the spectrometer and the mean value was obtained. For each element, the limit of quantification (*LOQ*) was calculated as follows:(5)$$LOQ=10\times \frac{SE}{b}$$
where *SE* = the standard error of the regression for each element and *b* = the value of the slope of the calibration curve for each element.

Vitamin B_12_ was quantified using a VitaFast^®^ Vitamin B_12_ microtiter plate test kit (R-Biopharm, Darmstadt, Germany) according to the manufacturer’s instructions and slight modifications as reported by Ssepuuya et al [31]. Briefly, vitamin B_12_ was extracted from 1.000 g of a sample using sodium cyanide and *taka diastase* (Sigma Aldrich, St. Louis, MO, USA) enzyme was added. To the selected wells of a microtiter plate, which were coated with *Lactobacillus delbrueckii subsp. lactis (leichmannii)*, the vitamin B_12_ assay medium was added followed by either the standard solution, sample, or blank. Incubation of the plates took place in the dark at 37 °C for 48 h. The intensity of metabolism or growth of *Lactobacillus delbrueckii* in reaction to the extracted vitamin B_12_ was measured as turbidity at 630 nm using a microtiter plate reader (VERSAmax EXTR, Molecular Devices, San Jose, CA, USA). The vitamin B_12_ content was then calculated by comparison to a standard curve using the SoftMax^®^ Pro 5 software and expressed in µg/100g dry mass.

### 2.4. Predicted Nutritional Quality Parameters

#### 2.4.1. Protein Quality

Values obtained from the amino acid profiles were used to predict the following protein quality indicators:

##### Predicted Protein Efficiency Ratio

The Predicted Protein Efficiency Ratio (Predicted *PER*-value) was calculated according to Alsmeyer et al. 1974 [26] using the following formula:(6)$$PER_{predicted}=-0.468+0.454\times \left(\xi_{Leu})-0.105\times (\xi_{Tyr}\right)-0.1539$$
where *ξ* = mass ratios of leucine (*Leu*) and tyrosine (*Tyr*) on insect protein (g/16 g N).

##### Essential Amino Acid Index

In order to estimate the Essential Amino Acid index (*EAAI*), the formula below, developed by Oser et al. 1959 [25], was used.
(7)$$EAAI=\sqrt[{9}]{\frac{\xi_{Lys\left(i\right)}}{\xi_{Lys\left(r\right)}}\times \frac{\xi_{Trp\left(i\right)}}{\xi_{Trp\left(r\right)}}\times \frac{\xi_{Ile\left(i\right)}}{\xi_{Ile\left(r\right)}}\times \frac{\xi_{Val\left(i\right)}}{\xi_{Val\left(r\right)}}\times \frac{\xi_{Arg\left(i\right)}}{\xi_{Arg\left(r\right)}}\times \frac{\xi_{Thr\left(i\right)}}{\xi_{Thr\left(r\right)}}\times \frac{\xi_{Leu\left(i\right)}}{\xi_{Leu\left(r\right)}}\times \frac{\xi_{Phe+Tyr\left(i\right)}}{\xi_{Phe+Tyr\left(r\right)}}\times \frac{\xi_{Met+Cys\left(i\right)}}{\xi_{Met+Cys\left(r\right)}}}$$
where $\xi_{AA\left(i\right)}$= mass ratio of insect amino acid on insect protein (mg/100 g protein) and $\xi_{AA\left(r\right)}$= mass ratio of reference amino acid on reference protein (mg/100 g protein).

Egg protein was used as a reference.

##### Predicted Biological Value

The Predicted Biological Value (*BV*) is a measure of how efficiently a dietary protein source is transformed into body tissue. To estimate the *BV* (predicted) using essential amino acid indices, the predicted biological value is obtained by the linear relationship with respect to the *EAAI* [25]:(8)$$BV_{predicted}=\left(1.09\times EAAI-11.7\right)$$

##### Nutritional Index

Another protein quality parameter is the Nutritional Index (*NI*, in%) that associates *EAAI* to the percent crude protein. The *NI* was calculated for all samples, according to Oser et al. (1959) [25]:(9)$$NI=\frac{EAAI\times w_{protein}}{100}$$

#### 2.4.2. Lipid Quality

To evaluate the lipid quality of the oils from these insects, values obtained from the fatty acid profiles were used to compute the following quality indices:

Atherogenic Index.

The Atherogenic Index (*AI*) is an index that can be used to estimate the risks for cardiac disorders. Its calculation is based on the content of the fatty acids of lauric acid (C12:0), myristic acid (C14:0), and palmitic acid (C16:0) and the groups monounsaturated fatty acid (*MUFA*) and polyunsaturated fatty acid (*PUFA*) [32]:(10)$$AI=\frac{w_{C12:0}+4\times w_{C14:0}+w_{C16:0}}{w_{MUFA}+w_{PUFA}}$$
where *w* = mass fraction of fatty acid(s) on total fatty acid content (mg/100 g fatty acids).

Polyunsaturated/Saturated Fatty Acid Ratio.

This is the ratio of the total polyunsaturated fatty acids (*PUFA*) content to that of the total saturated fatty acids (*SFA*) content or *P*/*S*:(11)$$P/S=\frac{w_{PUFA}}{w_{SFA}}$$

Thrombogenic Index.

The thrombogenic index is a ratio that estimates the thrombogenicity (clot-forming ability) of a food (oil) as proposed by Ulbricht and Southgate (1991) [32].
(12)$$TI=\frac{w_{C14:0}+w_{C16:0}+w_{C18:0}}{0.5\times w_{MUFA}+0.5\times w_{PUFA-n6}+3\times w_{PUFA-n3}+\frac{w_{PUFA-n3}}{w_{PUFA-n6}}}$$
where *w* = mass fraction of fatty acid(s) on total fatty acid content (g/100 g fatty acids), *TI* = thrombogenic index, *MUFA* = monounsaturated fatty acids, *SFA* = saturated fatty acids, *PUFA* = poly-unsaturated fatty acids, *UFA* = unsaturated fatty acids, *n*6 = omega-6, and *n*3 = omega-3 fatty acids.

### 2.5. Statistical Analyses

Data of proximate, amino acid, fatty acid, mineral composition, and vitamin B_12_ contents from the insect samples were subjected to Analysis of Variance (ANOVA) after testing for normality by means of a *Shapiro–Wilk* test using GraphPad Prism (GraphPad Software, La Jolla, CA, USA) version 8.4.3 for Windows. The difference between means was separated using the Tukey post hoc test at 5% (*p* = 0.05). The results are reported as means ± standard deviations of three technical replicates obtained from pooled samples. The standard deviation formulae for sums (*NFE*) and products (energy) were estimated using the equations derived by B.R. Scott [33].

## 3. Results

### 3.1. Proximate Composition

The results for moisture content, crude fat, crude protein, chitin (fibre), and ash are shown in Table 2. Proteins and fat constituted the most abundant macronutrients, accounting for 80–86% of total nutritional components. Protein content was significantly higher (*p* < 0.0001) in *G. bimaculatus* than in either locust species, for which *S. gregaria* contained significantly more protein than *L. migratoria* (*p* < 0.001). Fat content was significantly higher in *L. migratoria* (*p* < 0.001) than in either *G. bimaculatus* or *S. gregaria*, which had comparable fat contents. The fibre content of *L. migratoria* was significantly higher than that of *G. bimaculatus* and *S. gregaria*, which were again similar. The ash contents of the three orthopteran species showed no significant differences. The available carbohydrate content expressed as a nitrogen-free extract (*NFE*) was significantly higher (*p* < 0.0001) in *S. gregaria* compared to that of *G. bimaculatus* and *L. migratoria*, and *NFE* did not significantly differ between these latter two species.

### 3.2. Amino Acid Content

The amino acid profiles (in mg/100 g protein) of *G. bimaculatus*, *L. migratoria*, and *S. gregaria* are shown in Table 3. Glutamic acid was the most abundant amino acid recorded for *G. bimaculatus* (131 mg/100 g), *L. migratoria* (123 mg/100 g), and *S. gregaria* (120 mg/100 g). Methionine is the least abundant amino acid in both *G. bimaculatus* (0.70 mg/100 g) and *S. gregaria* (3.90 mg/100 g). The highest total amino acid content was found in *S. gregaria*, followed by *G. bimaculatus* and *L. migratoria*. Individual amino acid concentrations varied widely across the three orthopteran species, with the exceptions of histidine, threonine, cysteine, isoleucine, leucine, and tryptophan. The differences in the concentrations of serine (*p* < 0.0001), arginine, (*p* < 0.0001), glutamic acid (*p* < 0.0001), alanine (*p* < 0.0001), and methionine (*p* = 0.0012) between the cricket (*G. bimaculatus*) and the two locusts species were particularly high.

The ratio of essential to nonessential amino acids was reasonably constant (0.36–0.37) and did not vary among the three species. The Essential Amino Acid Index (*EAAI*) and Nutritional Index (NI) were significantly different at α = 0.05 for all three species. The highest *EAAI* and *NI* values were recorded for *S. gregaria* (76.0 and 46.7%), while *G. bimaculatus* had the lowest values. By contrast, *L. migratoria* had the lowest *NI* (39.5%). The biological value (*BV*) of the three species ranged between 63.7–71.3% and varied significantly.

### 3.3. Fatty Acid Content

The fatty acid compositions of *G. bimaculatus*, *L. migratoria*, and *S. gregaria* are presented in Table 4. The predominant fatty acids in all three species were palmitic and oleic acids, while eicosapentaenoic acid (EPA) was the least abundant. Linoleic acid was the most abundant polyunsaturated fatty acid in all three species. Of the monounsaturated fatty acids, oleic acid had the highest concentration in *S. gregaria* and *G. bimaculatus* while myristoleic acid had the lowest concentrations in both species. Myristic and lauric acid were low in all three species.

### 3.4. Mineral Composition and Vitamin B_12_ Content

The mineral compositions of the three orthopteran species are shown in Table 5. There were significant differences (*p* < 0.001) observed for all the mineral elements tested in all three species. The highest concentration of potassium was recorded in *S. gregaria*, then *G. bimaculatus*, and finally *L. migratoria*. Phosphorus, the next abundant mineral, was highest in *S. gregaria* and least in *L. migratoria.* Sodium was highest in *G. bimaculatus*, followed by *S. gregaria*, and least in *L. migratoria*.

Among the three least abundant elements were copper (significantly higher in *S. gregaria*) and then manganese (which recorded significantly higher amounts in *G. bimaculatus*). The ratio of calcium to phosphorus (Ca/P) (0.08–0.22) did not differ significantly among the three species. Amongst the micro minerals, zinc was the most abundant (12.7–24.9 mg/100 g), and the least in all three species was selenium (0.033–0.125 mg/100 g).

The vitamin B_12_ (cobalamin) content for the three species is shown in Table 5. Vitamin B_12_ levels in the three orthopterans varied significantly, with the highest values recorded in *G. bimaculatus*.

## 4. Discussion

Despite the economic importance of locusts, very little research has directly compared their nutritional profile to that of the more commonly reared and eaten crickets like *G. bimaculatus*, which are frequently used in powdered form as a food ingredient. We have demonstrated that powdered preparations of both locust species are of comparable nutritional quality to that of *G. bimaculatus*. Also, given the widespread consumption across sub-Saharan Africa and the display in swarming behaviour to another orthopteran, *Ruspolia differens (R. differens)*, nutritional comparisons are frequently made throughout the discussion between the latter and the insects of the present study. The contrast is further established because all four insects (*L. migratoria*, *S. gregaria G. bimaculatus*, and *R. differens*) were subjected to the same processing, analyses, and measuring equipment. The simplicity of the processing outlined in this paper, from washing to blanching, then oven-drying, and blending to powder, exemplifies their ease for inclusions and more frequent use as a food ingredient or dietary supplement. Therefore, this paper also aims to strengthen advocacy for the use of such minimally processed insect powders as processed food ingredients that could affordably be incorporated into other food products [34,35,36]. The current literature suggests evidence to support that oven-dried insects are comparable to freeze-dried ones. At the same time, oven-drying reduces energy input costs and improves specific physicochemical attributes such as reducing lipid oxidation and improving protein solubility [23,37,38]. Additionally, the nutritional value of the cricket and locust proteins after oven-drying at higher temperatures is high, as was also the case for mealworms and lesser mealworms as reported by other studies [39].

The protein content of edible insect species is high, ranging from 35% (in termites) up to 77% (in grasshoppers) [20,40], a range which is consistent with that of the current findings. However, the crude protein content in *L. migratoria*, *S. gregaria*, and *G. bimaculatus* are significantly higher than that of *R. differens* (47.7%) which was measured in the same study [23]. Our findings agree with that of previous studies that reported that the crude protein levels of all orthopterans ranged between 43.9–77.1% [41,42,43].

The protein values reported in *L. migratoria*, *S. gregaria*, and *G. bimaculatus* are superior or comparable to most of the conventional protein sources, for instance, beef consumed in Kenya [42]. Comparing the protein values (on dry basis) of these insects to that of Kenya beef (76.60%), all three insect flours had lower values to that of beef in all protein amounts. These comparable values are an indication that these insect flours can serve as an alternative protein source to conventional beef proteins.

The lipid content of the locusts and cricket analysed in the present study were quite variable and significantly lower than the values reported for *R. differens* (35.60%) [23], though slightly higher than the averages reported for other orthopterans (13.41%) [3]. The high lipid content of *R. differens* can be attributed to their highly polyphagous nature and their ability to store higher quantities of lipids during the early stages of life [44]. However, *L. migratoria* in this study showed higher quantities of lipids to that of soybean (a commercial lipid source), with approximately 20% lipids on a dry weight basis [45]. Thus, the locusts *L. migratoria* and *S. gregaria* could serve as alternative sources of oils after defatting and can be used as substitutes for the expensive fish oil in animal feeds or human food [36]. The high fat content reported for orthopterans is a direct indication of energy, and the energy values recorded in the current study are within the range reported in the literature for other species (400–500 Kcal/g) [3]. This implies that the consumption or inclusion of these insect meals in food products would contribute to meeting energy requirements in human and animal nutrition [46].

The crude fibre contents of *G. bimaculatus*, *L. migratoria*, and *S. gregaria* are lower compared to that of *R. differens* (Table 2) [23]. These crude fibre results are similar to that reported in the literature for other orthopterans (1–22%) [41,43]. Studies have shown that adequate fibre intake confers some health benefits such as lowering serum cholesterol level, the risk of coronary heart diseases, hypertension, constipation, and diabetes [47]. The high fibre content of *L. migratoria* is comparable to that of *R. differens* and might therefore confer greater health benefits for humans and animals. The crude fibre in insects is principally composed of chitin [48], which is a raw material frequently used in the industrial production of chitosan, oligosaccharides, and glucosamine [48].

The ash contents of *G. bimaculatus* and *S. gregaria* were comparable to that of *R. differens* (4.66%) [23]. Extremely low ash contents have been reported in other orthopterans such as *Sphenarium mexicanum* with 0.34% [43]. It is important to note that the ash content reported for the three species in this study is consistent with previous findings for orthoptera (0.34–9.36%) [43].

In the present study, the three orthopteran species had all the essential amino acids (except methionine) in adequate amounts required for human and animal nutrition [46]. Nutritionally, protein-based food materials are of acceptable quality when biological values are above 70% and the essential amino acid index is higher than 0.70 [49]. *PER* is the ratio of the weight gained of a test group or sample to the total proteins consumed [49]. *PER*
_predicted_ is a calculated parameter that accurately estimates the real or actual *PER* values [26]. A *PER* below 1.5 connotes a low-quality protein, that between 1.5–2.0 has an intermediate quality, and that above 2.0 is considered adequate to high quality [49]. As such, the following familiar protein sources are widely considered as high quality: casein (2.5); beef muscle, fish, and poultry (2.7); and eggs (3.1) [50]. The *PER*
_predicted_ values calculated from the amino acid compositions of the insects in our study (2.3–2.4) were above the 2.0 threshold and compared favourably with the above traditional protein sources and therefore can be classified too as high-quality proteins. The mean *EAAI* of the blanched and oven-dried *L. migratoria*, *S. gregaria*, and *G. bimaculatus* is comparable to that of the chicken egg, estimated to be 1.0 [25]. Apart from *G. bimaculatus*, the *EAAI* of *L. migratoria* and *S. gregaria* were higher than 70%, which is similar to that of *R. differens* (72%) [23], thus confirming the high protein nature of the insects. The calculated *EAAI*s of *L. migratoria* and *S. gregaria* are higher than those reported for soybean (65–72%) [25]. Thus, the *EAAI* values recorded in the present study meet the standard requirements for adequate human nutrition [24]. The lysine and threonine levels in *L. migratoria* and *S. gregaria* were of reasonable quantities compared to predominantly cereal-based diets (3.7–4.2 mg/100 g and 3.2–3.4 mg/100 g protein, respectively) of most African rural communities [51]. This could indicate that the inclusion of locust meals in food products in plague-affected regions would have a positive impact on the household nutritional status of the various communities.

One limitation of this study is the lack of protein digestibility data, which could predict the overall efficiency of protein utilization using the Protein Digestibility-Corrected Amino Acid Score (PDCAAS). This evaluation method is composed of the Biological Value (*BV*) and the digestible value of proteins concerned. Studies on other insects have shown that industrial drying methods like oven-drying have minimal influence on PDCAAS [39]. The FAO/WHO [52] recommended PDCAAS as the standard protein quality evaluation method, but in 2013, they endorsed a new evaluation method, the Digestible Indispensable Amino Acid Score (DIAAS). Therefore, additional studies that can relate predicted biological values and protein efficiency ratios to either the PDCAAS and or DIAAS would be of more relevance to human nutrition.

Lauric acid (C12:0) and myristic acid (C14:0) were present in the three orthopteran species in minute amounts. The presence of both fatty acids has been implicated in raising harmful cholesterol levels in the blood serum [53]. Therefore, being present in small amounts, the inclusion of these insects in foods would not create any adverse cholesterol implications. In contrast to the abovementioned saturated fatty acids, the inclusion of poly- or mono-unsaturated fatty acids such as oleic acid (C18:1), linoleic acid (C18:2), and α-linolenic acid (C18:3) in diets is recommended for preventing cardiovascular diseases [54]. The presence of linoleic and α-linolenic acid as essential polyunsaturated fatty acids in all three orthopteran species was remarkable. These fatty acids are crucial in the development of children under five years of age and in women of reproductive age [53]. For optimal growth and development of children’s brains, polyunsaturated acids such eicosatetraenoic acid (20:4) and docosahexaenoic acid (22:6), which are synthesised from their respective essential fatty acid precursors, linoleic acid (18:2) and α-linolenic acid (18:3), are required [53]. Deficiency in some essential fatty acids in children can lead to malnourishment and several clinical problems such as the increased risk of infection, impaired wound healing, fatty liver, and psychomotor changes coupled with growth retardation [32,53,55].

The nutritional qualities of *G. bimaculatus*, *L. migratoria*, and *S. gregaria* were demonstrated by the presence of saturated (*SFA*), monounsaturated (MFA), and polyunsaturated fatty acids (*PUFA*) *n* − 6 *PUFA*, *n* − 3 *PUFA;* omega 6 to omega 3 ratios (*n*6/*n*3); and saturation (*P*/*S*), atherogenic (*AI*), and thrombogenic (*TI*) indexes [32]. *G. bimaculatus* exhibited a very high *n*6/*n*3 ratio compared to *L. migratoria* and *S. gregaria* lipids that had negligible levels of *n*6/*n*3. A good *n*6/*n*3 ratio lies between 1:1–1:6; however, there is contention regarding these ratios among researchers [56]. This would suggest that consumption of the fats from these two locusts would have very positive effects on human cardiovascular health. Furthermore, studies have suggested that a high *n*6/*n*3 ratios (>20:1) in the diet might be linked to the development of a variety of physiological disorders (such as cancer, coronary heart disease, etc.) [56]. The polyunsaturated to saturated fatty acid (*P*/*S*) ratio is one of the most significant markers of lipid composition in a healthy diet. It is recommended to consume a diet with a P/S ratio close to 1. A high P/S ratio (≥3) in the diet may promote tumour formation, while a low P/S ratio (≤0.33) in the diet could be atherogenic [57]. Interestingly, *G. bimaculatus* lipids had a P/S closest to 1 (0.76), which means it contains more of the “desirable” polyunsaturated than the “less desirable” saturated fatty acids. At the same time, the locusts performed poorly in this parameter, though less than 0.33, which is the cut-off mark to be considered atherogenic. Ulbricht and Southgate [32] proposed the term “atherogenic index” (*AI*) for lipids as a nutritional index for the risk of cardiovascular diseases. The *AI* index was obtained using the Table 4 values for lauric (C12:0), myristic (C14:0), and palmitic (C16:0) acids and unsaturated fatty acids. An increase in the *AI* index increases the risk of the incidence of cardiovascular diseases in humans. The *AI*s of the diets of the Eskimo (0.39), British (0.93), and Danish (1.29) [32] suggest that consuming 100 g dry weight of any of the examined insects (except for *L.migratoria*) would fall within the recommended value for these countries. Additionally, our study revealed atherogenic index values for *G. bimaculatus* and *S. gregaria*, which were significantly lower and compared favourably with those for other animals such as sheep (0.7–0.9), beef (0.7), pork (0.6), and poultry (0.5) [58,59,60]. Consequently, the consumption of food products composed of *G. bimaculatus* and *S. gregaria* with lower atherogenic index values may lead to a decrease in the total cholesterol and the LDL cholesterol in human blood plasma. Additionally, the lower atherogenic index observed in our study highlights that *G. bimaculatus* and *S. gregaria* had low concentrations of the saturated fatty acids lauric (C12:0), myristic (C14:0), and palmitic (C16:0) acids in comparison to other lipid sources [54]. The high risk of cardiovascular diseases in humans warrants a more thorough look into the fatty acids of some insects as a possible replacement of other lipid sources [32].

Minerals play vital roles in numerous biological processes, and in many developing nations, micronutrient deficiencies are still widespread. Such micronutrient deficits, especially for trace minerals, can have severe health outcomes affecting growth and development. In this study, the high potassium values recorded for the three orthopteran species are in agreement with other related studies (724–834 mg/100 g DM) [3,23,61], which can be attributed to the food plants known to contain high levels of potassium [62]. The phosphorus contents of *G. bimaculatus*, *L. migratoria*, and *S. gregaria* were high, thus influencing the calcium–phosphorus ratio, which was less than one. The phosphorus in most insects is readily available, as shown for *Musca autumnalis* puparia with 92% availability [62]. *S. gregaria* had the highest amounts of Mg, Zn, Fe, K, and Cu when compared to *G. bimaculatus* and *L. migratoria*. The Ca/P ratio of *G. bimaculatus*, *L. migratoria*, and *S. gregaria* was within the recommended range between 0.1–2.0 for this ratio. The Ca/P ratio is a tool to indicate facilitation of bone and teeth formation in children, and this ratio also signals the incidence of osteoporosis in adults [63]. This implies that consumption of any of the orthopteran species might help to improve bone and teeth formation in infants and to reduce osteoporosis in adults [63].

The vitamin B_12_ content observed in *G. bimaculatus*, *L. migratoria*, and *S. gregaria* is comparable to that reported for *R. differens (*0.88–1.35 µg/100 g) [23] but much less than that reported for *Acheta domesticus* adults (5.4 µg/100 g) and nymphs (8.7 µg/100 g) by Van Huis et al. 2013 [7]. When compared to other animal-based food sources, the vitamin B_12_ content recorded in the three orthopteran species were, however, lower than that documented for pork and poultry meat (9.3 µg/100 g) [64]. This implies that the consumption of *G. bimaculatus*, *L. migratoria*, and *S. gregaria* would be capable of providing an adequate amount of vitamin B_12_, known to play a useful role in DNA synthesis, in regenerating methionine for protein synthesis and methylation, and in preventing homocysteine accumulation [65,66]. The recommended dietary allowance (in mg/day) of vitamin B_12_ is 0.4 mg for the first six months of life, 0.5 mg for 6–12 mo., 0.9 mg for 1–3 years, 1.2 mg for 4–8 years, 1.8 mg for 9–13 years, and 2.4 mg for 14 years through old age. In pregnancy, 2.6 mg is recommended, and in lactation, 2.8 mg is recommended [65]. Thus, vitamin B_12_ fortification of flours with that of these insects is highly recommended in developing countries, especially among those who decisively evade all animal source foods [66].

## 5. Conclusions

We can conclude that the three orthopteran species, *G. bimaculatus*, *L. migratoria*, and *S. gregaria*, can be considered as potential sustainable, high-quality food sources, given that the protein values are comparable or superior to those of most plants and animal sources commonly consumed as foods. For the first time, the high quality of insect proteins was confirmed with predicted protein quality parameters such as *PER* and *BV*. Additionally, their high mineral content, good amino acid quality, suitable fatty acid quality, and high vitamin B_12_ content make them a worthy replacement or substitute for meat. Our findings revealed that the inclusion of these insect meals in food products might help to mitigate most of the energy and protein insufficiency issues observed in most sub-Saharan African countries. The highlight of this study is to promote the use of dried and processed insects instead of highly perishable fresh insects.

Furthermore, the blanched and oven-dried insect powder can be incorporated into other foodstuffs, enhancing nutritional value in this way. Thus, by minimally processing these insects into their flours post-harvesting through simple processes including blanching, oven-drying, and blending, they provide a nutrient-rich and affordable food ingredient. Of these three insects examined, we recommend that *S. gregaria* should be the most consumed since it fared better in nutritional attributes and is the most devastating locust pests that could save a lot on insecticides. These insects can be easily mass-harvested from the wild, implying that incorporating them into local diets would help to combat food and nutritional insecurity; would assist with job creation to generate much-needed household income, and would improve livelihoods. Therefore, in the event of natural swarms, we recommend the safe harvesting and processing of locusts for food and feed as an integrated approach for management of these ravaging pests in affected areas.

## Figures and Tables

**Table 1 foods-10-00144-t001:** Inductively coupled plasma optical emission spectrometry (ICP-OES) parameter settings.

Sampler		Spectrometer	
Parameter	Setting	Parameter	Setting
Plasma conditions	Same for each element	Pulsed gas flow	Normal
Type Aerosol	Wet	Spectral profiling	No
Start nebulizer	Directly	Resolution	Fixed (normal)
Sample flow (mL/min)	1.5	Reading time (s)	Automatic
Plasma sight (all)	Radial	Break time (s)	30
Plasma sight (element)	Axial	Replications (#)	3
Source delay (s)	30	Software	WinLab 32
Flush time (s)	10		

Elemental limits of quantification (*LOQ*) in ppm: Na: 4.97; Ca: 2.16; K: 2.63; Mg: 4.83; Zn: 0.14; Fe: 0.22; P: 5.94 Cu: 0.02; Mn: 0.05.# = Number

**Table 2 foods-10-00144-t002:** Proximate composition of oven-dried *G. bimaculatus*, *L. migratoria*, and *S. gregaria* (based on a dry matter basis except for the moisture content of fresh samples): each value represents the mean ± standard deviation of triplicate determinations (except for moisture contents, *n* = 5).

Parameter	*G. bimaculatus*	*L. migratoria*	*S. gregaria*
Moisture (fresh)	73.66 ± 2.27	77.77 ± 1.13	82.39 ± 2.20
Moisture (processed)	0.85 ± 0.16 ^a^	1.46 ± 0.16 ^b^	1.97 ± 0.18 ^c^
Protein	65.34 ± 0.48 ^a^	54.16± 0.93 ^b^	61.41± 0.32 ^c^
Fat (total lipids)	20.74 ± 0.16 ^a^	30.52 ± 0.75 ^b^	19.10 ± 0.10 ^a^
Fibre (chitin)	5.80 ± 1.45 ^a^	9.19 ± 0.32 ^b^	6.61 ± 1.28 ^a^
Ash	4.11 ± 0.01 ^a^	3.08 ± 0.06 ^a^	2.70 ± 0.13 ^a^
*NFE*	4.80 ± 1.53 ^a^	1.50 ± 1.25 ^a^	6.50 ± 1.34 ^b^
Energy (Kcal/g)	469.91 ± 44.09	512.34 ± 43.36	474.76 ± 47.40

*NFE* = nitrogen-free extract. ^a^, ^b^, ^c^ Mean values in the same row with a different letter superscript are significantly different (*p* < 0.05).

**Table 3 foods-10-00144-t003:** Amino acid profile (in mg/100 g protein), recommended daily intake, and predicted protein quality indicators of *G. bimaculatus*, *L. migratoria*, and *S. gregaria*: each value represents the mean ± standard deviation of triplicate determinations.

Amino Acid	*G. bimaculatus*	*L. migratoria*	*S. gregaria*	Recommended Daily Intakes (WHO/FAO) **
Histidine	25.79 ± 0.51 ^a^	27.23 ± 1.40 ^a^	26.32 ± 0.93 ^a^	15.0
Serine	55.84 ± 0.34 ^a^	40.71 ± 0.76 ^c^	44.73 ± 3.29 ^b^	----
Arginine	74.21 ± 1.32 ^a^	59.12 ± 2.86 ^b^	58.32 ± 1.90 ^b^	----
Glycine	55.14 ± 1.10 ^a^	68.92 ± 2.47^b^	65.11 ± 1.33 ^c^	----
Aspartic acid	104.45 ± 1.52 ^a^	81.15 ± 1.70 ^b^	80.53 ± 1.56 ^b^	----
Glutamic acid	131.04 ± 2.86 ^a^	122.45 ± 1.19 ^b^	120.29 ± 2.05 ^b^	----
Threonine	41.31 ± 0.34 ^a^	39.62 ± 0.68 ^a^	39.47 ± 0.77 ^a^	23.0
Alanine	90.23 ± 3.84 ^b^	122.96 ± 1.17 ^a,b^	120.10 ± 2.05 ^a^	----
Proline	64.73 ± 1.68 ^a^	76.35 ± 0.96 ^a^	74.28 ± 0.13 ^a,b^	----
Cysteine	4.40 ± 0.11 ^a^	5.67 ± 1.71 ^a^	7.36 ± 0.23 ^a,b^	6.6
Lysine	57.20 ± 0.15 ^a^	47.84 ± 3.84^b^	45.69 ± 2.18 ^b^	45.0
Tyrosine	51.36 ± 1.36 ^c^	56.60 ± 4.88 ^b^	66.66 ± 3.07 ^a^	----
Methionine	0.70 ± 0.49 ^b^	3.91 ± 2.36 ^b^	9.49 ± 4.90 ^a^	16.0
Valine	64.59 ± 1.44 ^a,b^	72.11 ± 0.79 ^a^	68.68 ± 0.35 ^b^	39.0
Isoleucine	46.08 ± 0.13 ^a^	46.18 ± 0.29 ^a^	46.25 ± 0.82 ^a^	30.0
Leucine	83.47 ± 0.06 ^a^	84.56 ± 0.26 ^a^	82.30 ± 1.7 ^a^	59.0
Phenylalanine	38.90 ± 0.41 ^a^	34.9 ± 1.5 ^a^	36.3 ± 1.73 ^a^	----
Tryptophan	10.52 ± 0.76 ^a^	9.71 ± 1.42 ^a^	8.20 ± 0.41 ^a^	6.0
∑ Amino acids (%) *	55.93 ± 2.65 ^a^	47.74 ± 5.27 ^a^	58.46 ± 3.46 ^a^	
Essential (E)	368.57 ± 1.06 ^a^	366.06 ± 2.84 ^a^	362.65 ± 3.16 ^a^	
Non-Essential (N)	631.40 ± 1.08 ^a^	633.94 ± 2.88 ^a^	637.36 ± 3.19 ^a^	
E/N	0.58 ± 0.00 ^a^	0.58 ± 0.01 ^a^	0.57 ± 0.01 ^a^	
E + N	1000 ± 0.00 ^a^	1000 ± 0.00 ^a^	1000 ± 0.00 ^a^	
E/(E + N)	0.37 ± 0.00 ^a^	0.37 ± 0.00 ^a^	0.36 ± 0.00 ^a^	
*PER* _predicted_	2.32 ± 0.12 ^a^	2.39 ± 0.29 ^a^	2.42 ± 0.23 ^a^	
*EAAI*	69.23 ± 0.96 ^a^	73.00 ± 1.25 ^b^	76.10 ± 1.82 ^c^	
*BV* _predicted_	63.76 ± 1.05 ^a^	67.87 ± 1.36 ^b^	71.25 ± 1.98 ^c^	
Nutritional Index _predicted_	45.24 ± 0.94 ^b^	39.53 ± 0.23 ^a^	46.73 ± 1.11 ^c^	

^a^, ^b^, ^c^ Mean values in the same row with a different letter superscript are significantly different (*p* < 0.05). * Expressed as percent 100 g per insect sample. ** Food and Agriculture Organization (FAO)/WHO Amino acid requirements in humans (mg/100 g protein). --- data not available.

**Table 4 foods-10-00144-t004:** Fatty acid profile (in g/100 g fatty acid) of *G. bimaculatus*, *L. migratoria*, and *S. gregaria*: each experimental value represents the mean ± standard deviation of triplicate determinations.

Fatty Acid	Class	*G. bimaculatus*	*L. migratoria*	*S. gregaria*
Decanoic acid (C10:0)	*SFA*	0.10 ± 0.02 ^a^	1.08 ± 1.25 ^a^	0.15 ± 0.11 ^a^
Lauric acid (C12:0)	*SFA*	1.60 ± 0.09 ^a^	1.30 ± 1.17 ^a^	0.41 ± 0.06 ^a^
Myristic acid (C14:0)	*SFA*	0.74 ± 0.45 ^a^	3.63 ± 0.56 ^a^	2.66 ± 0.25 ^a^
Myristoleic acid (C14:1)	*MUFA*	0.10 ± 0.02 ^a^	1.25 ± 1.54 ^a^	0.15 ± 0.13 ^a^
Pentadecanoic acid (C15:0)	*SFA*	0.19 ± 0.09 ^a^	0.40 ± 0.35 ^a^	0.22 ± 0.18 ^a^
Palmitic acid (C16:0)	*SFA*	27.73 ± 2.42 ^a^	43.01 ± 1.45 ^b^	26.14 ± 1.21 ^a^
Palmitoleic acid (C16:1)	*MUFA*	1.05 ± 0.30 ^a^	1.60 ± 0.96 ^a^	1.34 ± 0.10 ^a^
Heptadecanoic acid (C17:0)	*SFA*	0.29 ± 0.02 ^a^	1.14 ± 0.83 ^a^	0.34 ± 0.13 ^a^
Stearic acid (C18:0)	*SFA*	7.36 ± 0.82 ^a^	6.31 ± 6.01 ^a^	7.02 ± 0.67 ^a^
Oleic acid (C18:1)	*MUFA*	31.76 ± 0.82 ^a^	22.85 ± 6.07 ^b^	40.87 ± 1.76 ^c^
Linoleic acid (C18:2) (*n*6)	*PUFA*	27.33 ± 0.41 ^a^	9.32 ± 0.90 ^b^	6.85 ± 0.18 ^b^
α-Linolenic acid (C18:3) (*n*3)	*PUFA*	1.02 ± 0.75 ^a^	4.85 ± 0.56 ^a^	13.11 ± 0.55^a^
Arachidonic acid (C20:4) )*n*6)	*PUFA*	0.59 ± 0.06 ^a^	1.40 ± 0.96 ^a^	0.43 ± 0.16 ^a^
Eicosapentaenoic acid (C20:5) (*n*3)	*PUFA*	0.13 ± 0.12 ^a^	1.89 ± 1.35 ^a^	0.29 ± 0.30 ^a^
Total Lipids		20.74 ± 0.16 ^b^	30.52 ± 0.75 ^c^	19.10 ± 0.10 ^a^
Total *SFA*		38.02 ± 1.15 ^a^	56.85 ± 4.64 ^b^	36.95 ± 1.43 ^a^
Total *MUFA*		32.91 ± 0.53 ^a^	25.70 ± 4.72 ^a^	42.37 ± 1.77 ^b^
Total *PUFA*		29.54 ± 0.40 ^b^	17.45 ± 3.54 ^a^	20.68 ± 1.77 ^a^
Total UFA		62.46 ± 0.34 ^b^	43.15 ± 4.64 ^a^	63.05 ± 1.89 ^b^
Total essential (C18:2 + C18:3)		28.82 ± 0.45 ^b^	14.17 ±1.23 ^a^	19.96 ± 0.58 ^a^
Total *n*6		27.92 ± 0.39 ^b^	10.71 ± 1.79 ^a^	7.28 ± 0.24 ^a^
Total *n*3		1.15 ± 0.86 ^a^	6.74 ± 1.83 ^b^	13.40 ± 0.63 ^c^
*n*6/*n*3 ratio		17.19 ± 0.58 ^c^	1.63 ± 0.26 ^b^	0.54 ± 0.01 ^a^
*PUFA*/*SFA* (*P*/*S*) ratio		0.78 ± 0.02 ^b^	0.31± 0.07 ^a^	0.56 ± 0.03 ^b^
Atherogenic index (*AI*)		0.48 ± 0.02 ^a^	1.06 ± 0.12 ^a^	0.44 ± 0.03 ^a^
Thrombogenic index (*TI*)		1.11 ± 0.07 ^a^	3.19 ± 0.62 ^a^	1.56 ± 0.10 ^a^

^a^, ^b^, ^c^ Mean values in the same row with a different letter superscript are significantly different (*p* < 0.05). *MUFA* = Monounsaturated fatty acids, *SFA* = Saturated fatty acids, *PUFA* = Poly-unsaturated fatty acids, *UFA* = Unsaturated fatty acids, *n*6 = omega-6, and *n*3 = omega-3 fatty acids.

**Table 5 foods-10-00144-t005:** Recommended daily intakes, mineral content (mg/100 g dry matter), and vitamin B12 content (µg/100 g dry matter) of *G. bimaculatus*, *L. migratoria*, and *S. gregaria*: each value represents the mean ± standard deviation of triplicate determinations.

Mineral or Vitamin B12	*G. bimaculatus*	*L. migratoria*	*S. gregaria*	Recommended Daily Intakes (WHO/FAO) **
Potassium	1025.14 ± 0.03 ^a^	796.01± 0.01 ^b^	1309.46 ± 0.66 ^c^	4700
Phosphorus	882.38 ± 0.09 ^a^	697.17± 0.02 ^b^	968.65 ± 0.51 ^c^	700
Sodium	383.25 ± 0.01 ^a^	221.89 ± 0.01 ^b^	285.14± 0.15 ^c^	1500
Calcium	191.16 ± 0.01 ^a^	129.79 ± 0.01 ^b^	80.48 ± 0.04 ^c^	1300
Magnesium	110.57 ± 0.01 ^a^	86.01 ± 0.01 ^b^	128.29 ± 0.07 ^c^	260
Zinc	13.76 ± 0.01 ^a^	12.70 ± 0.01 ^b^	24.88 ± 0.01 ^c^	3–14
Manganese	7.15 ± 0.01 ^a^	0.40 ± 0.01 ^b^	1.26 ± 0.01 ^c^	1.8–2.6
Iron	4.60 ± 0.01 ^a^	6.59 ± 0.01 ^b^	7.31 ± 0.01 ^c^	7.5–59
Copper	1.84 ± 0.01 ^a^	1.20 ± 0.01 ^b^	4.86 ± 0.01 ^c^	0.9–1.3
Ca/P	0.22 ± 0.09	0.19 ± 0.02	0.08 ± 0.52	
‡ Vitamin B12 ¥	1.35 ± 0.14 ^a^	1.10 ± 0.06 ^b^	0.22 ± 0.01 ^c^	0.40–1.8

^a^, ^b^, ^c^ Mean values in the same row with a different letter superscript are significantly different (*p* < 0.05). ¥ = value expressed in µg/100 g dry matter; ** FAO/WHO (values in mg/day except for selenium). ‡ FAO 2001 (human vitamin and mineral requirements (for infants and children)).

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
