# Peer review of "Affordable Processing of Edible Orthopterans Provides a Highly Nutritive Source of Food Ingredients"

_foods, 2021, doi:10.3390/foods10010144_

Round 1

Reviewer 1 Report

The paper is very interesting and is within the scope of the Journal. However, it contain some major and minor inaccuracies, which I have pointed below:

  1. Abstract - line 20 - please specify that the content of protein is per dry weight
  2. The Introduction part is to extensive. There is no need to describe so many species of insects, which can be used as food sources. Please describe the most important focusing on their nutritional value.
  3. Part 2.2. Sample preparation - how many insects were used in chemical determinations. They have been counted or converted to mass (ex. g)?
  4. lines 143-144 - please breifly describe how each parameter was determined. Statement that it was previously described in some other paper is not enough.
  5. line 170 - how the insects oil were prepared? There is nothing in the text regarding this issue.
  6. lines 179 - Mineral and trace elemental analysis - this part was very brifely described. Which elements were determined? How was the accuracy of the method checked? Was this method validated? If so, there should be at least a brief description, and basic parameters of the method should have been provided.
  7. Table 1 - after oven drying the samples still contains 0.85-1.97% of moisture? Did the Authors determined moisture content in fresh insects? It would be very interesting to know what is the protein content in fresh inscets, not in dryied, as it is obvious that the content of protein in dry matter would be very high.
  8. Table 4 - the same as above. It is obvious, that the content of various nutrients (including minerals) in dry weight will be high. But the most important thing, is the content in fresh weight, because all the data regarding nutritional value of various foods are expressed per fresh weight. So, what is the sense to compare nutrients content in dry matter of one food with content in fresh weight in some other food? Therefore I asked what is the content of water in fresh weight of the investigated insects?
  9. lines 485-487 - "This implies that the consumption of L. migratoria would be adequate to satisfy the recommended daily intake (RDI), as shown in Table 4." It is hard to agree with this statement, as RDI or RDA values reffer to the consumption of nutrients with all foods in a form in which they can be ingested, and the values provided by the Authors are referred to dried insects and not fresh. If the content in fresh insects would be high we could say that this food is a rich source of particular nutrients. However, there is nothing in the text on the content of water in fresh weight. I am aware that the insects are not consumed without any culinary preparation, but it is very important to know what is the content of moisure in fresh insects to evaluate their nutritional value.

Reviewer 2 Report

-The authors describe a straightforward yet meaningful analysis of insect composition.

-The introduction adequately presents the rationale for the research. Minor editing and organization improvements are recommended.

-Were standards used for amino acid analysis? How many replicates were performed? Please clarify

-It would be more appropriate to report the composition of Kenya beef in the discussion rather than putting it in the results, which suggests that an actual analysis of Kenya beef was performed. Actually conducting the analysis would strengthen the comparison of this paper. Otherwise, please remove it from the results section (or at a minimum more clearly indicate that it is a reference).

-Because the PER is more relevant during growth states (such as children), a discussion of how your findings may compare with DIAAS (or even PDCAAS) would provide greater relevance. FAO currently recommends DIAAS (http://www.fao.org/documents/card/en/c/ab5c9fca-dd15-58e0-93a8-d71e028c8282/) . A discussion. Please update. Please also make sure protein requirement sources are current - ref 66 is from 1985. Other sources about protein quality are also more than 20 years old and should reflect more recent literature about the dietary relevance of protein quality. Otherwise, please limit the discussion of protein quality to avoid over-interpreting the results.

-Line 349. Sentence fragment.  

-Line 399 states that all AA are present at recommended levels, yet the table shows many that are lower than WHO requirements. Please clarify.

-Given that the rationale for this research is the ease of preparation and processing, a paragraph comparing nutritional quality vs freeze-drying would be of interest.

Author Response

See attachment provided, please.

Reviewer 3 Report

This work I believe that the work is in agreement with the Aims & Scope of the SpeJournal. It presents and highlights an interesting topic. However some corrections shall be implemented.

General comments

  1. The authors shall check font styles and follow the manuscript structure (template) of Foods.
  2. In my opinion the introduction part is very extended and shall be reduced to a smaller size, focusing and aiming to main points.
  3. Check fonts of Tables' captions and footnotes.
  4. Please add the following reference and discuss accordingly.

L.D. Jensen et al. Journal of Insects as Food and Feed: 5 (4)- Pages: 257 – 266, Nutritional evaluation of common (Tenebrio molitor) and lesser (Alphitobius diaperinus) mealworms in rats and processing effect on the lesser mealworm

5. Tables (all): In my opinion 1 decimal point is sufficient than 2, and shall be revised.

Specific comments

L.159: Only 6M HCl? No phenol or thioglycolic acid? Why? Explain and discuss in the text (results, discussion).

L.159-166: Please provide time and temperature conditions of the acidic hydrolysis.

L.159-166: Analysis with HPLC-PDA [44], and also describe the sample preparation. It is not clear if there is derivatization or not. Please be more descriptive and specific.

L.335 and L.343: Food additive? Please add respective reference. 

L.333-346: This paragraph requires revision and addition of supportive references. Revision and amendments are needed.

L.347-363: This paragraph is not referring to any results and it is not a discussion. It shall be removed or shorten and placed to support results (as par of the discussion).

Author Response

See attachment provided, please.

Round 2

Reviewer 1 Report

  1. I do not understand why the Authors put operation parameters in Table 1. These are just normal operation settings used during spectrometric analysis. Of course if the Authors want they can stay like this, however in my previous review I asked: "How was the accuracy of the method checked? Was this method validated?" I have expected Authors to put values like LOD, LOQ or recovery for each element to prove that the method used was appropriate for the investigated material. 
  2. lines 421-422 - the Authors wrote: "All samples were analyzed in duplicate". However, in the Table 1 there is "Replicas (#) 3"? So finally how many samples were determined? Another thing is, that in my opinion it is better to write replicates or repetitions, rather than "replicas", as during the analysis replicates of the same material were analyzed and not copies. 
  3. The Authors in their responses wrote: "Also, the aim and novelty of this study is to promote the use of dried and processed because of their extended shelf life compared to the highly perishable fresh insects. Furthermore, blanching and dying also improve safety. Afterwards the dried powder can be incorporated into other foodstuffs, enhancing the nutritional value in this way". Please put this statement (of course after appropriate editing) into Conclusion part, which will additionally emphasize the novelty of this study.  

Author Response

Dear Reviewer,

Please see the attached document with our responses.

Kind regards